# Antiparasitic Potential of Chromatographic Fractions of *Nephrolepis biserrata* and Liquid Chromatography-Quadrupole Time-of-Flight-Mass Spectrometry Analysis

**DOI:** 10.3390/molecules26020499

**Published:** 2021-01-19

**Authors:** Muhammad Dawood Shah, Kazuki Tani, Yoong Soon Yong, Fui Fui Ching, Sitti Raehanah Muhamad Shaleh, Charles S. Vairappan, Balu Alagar Venmathi Maran

**Affiliations:** 1Borneo Marine Research Institute, Universiti Malaysia Sabah, Kota Kinabalu, Jalan UMS 88450, Sabah, Malaysia; dawoodshah@ums.edu.my (M.D.S.); cfuifui@ums.edu.my (F.F.C.); sittirae@ums.edu.my (S.R.M.S.); 2Laboratory of Natural Products Chemistry, Institute for Tropical Biology and Conservation, Universiti Malaysia Sabah, Kota Kinabalu, Jalan UMS 88450, Sabah, Malaysia; tani_kindaisuiken@yahoo.co.jp (K.T.); csv@ums.edu.my (C.S.V.); 3Laboratory Centre, Xiamen University Malaysia, Jalan Sunsuria, Bandar Sunsuria, Sepang 43900, Selangor, Malaysia; yoongsoon.yong@xmu.edu.my

**Keywords:** natural products, aquaculture, grouper, *Zeylanicobdella*, antiparasitic, secondary metabolites, leeches, LC-QTOF-MS

## Abstract

Marine aquaculture development is recently impeded by parasitic leech *Zeylanicobdella*
*arugamensis* (Hirudinea, Piscicolidae) in Sabah, Malaysia. The parasitic leech infests a variety of cultured fishes in aquaculture facilities. In this study, we evaluated the antiparasitic activity of the chromatographic fractions of the medicinal plant *Nephrolepis biserrata* methanol extract against *Z.*
*arugamensis* and highlighted the potential metabolites responsible for the antiparasitic properties through liquid chromatography (LC)–quadrupole time-of-flight (QTOF)–mass spectrometry (MS) analysis. Out of seven fractions obtained through flash column chromatography techniques, three fractions demonstrated antiparasitic properties. Significant parasitic mortality was indicated by fraction 3 at a concentration of 2.50 mg/mL, all the leeches were killed in a time limit of 1.92 ± 0.59 min. followed by fraction 4 (14 mg/mL) in 34.57 ± 3.39 and fraction 5 (15.3 mg/mL) in 36.82 ± 4.53 min. LC-QTOF-MS analysis indicated the presence of secondary metabolites including phytosphingosine (**6**), pyrethrosin (**1**), haplophytine (**9**), ivalin (**2**), warburganal (**3**), isodomedin (**4**) and pheophorbide a (**16**), representing sphingoid, alkaloid, terpenoid, phenolic and flavonoid groups. Thus, our study indicated that the chromatographic fractions of *N. biserrata* demonstrated significant antiparasitic activity against the marine parasitic leeches due to the presence of potent antiparasitic bioactive compounds.

## 1. Introduction

The high demand for the fish product and decrement of ocean fisheries due to overfishing and deterioration of coastal environments which have triggered the rapid development of aquaculture facilities in Malaysia and other parts of the world, lead to the expansion of brackish-water and marine aquaculture [1,2]. Accordingly, the reputation of parasites as disease-causing agents in aquaculture has become evident [3,4]. Fish parasites are not only responsible for commercial losses in aquaculture and fisheries industries but also cause socio-economic implications both in developing and developed countries [5,6]. Different types of parasites including leeches (Annelida), monogeneans (Platyhelminthes), caligids (Crustacea), etc., are affecting various species of fishes [7,8,9,10].

Among the marine parasitic leeches, *Zeylanicobdella arugamensis* de Silva, 1963 (Hirudinea, Piscicolidae) is commonly found in Malaysia, Indonesia, Brunei, Philippines, Japan and other South-East Asian countries [3,4,7,11,12,13]. The leech is characterized by having an asymmetrical flattened cylindrical body with anterior and posterior suckers and has been considered to be the most harmful ectoparasite affecting a large variety of species of fish [3,7]. The mortality of the host fish typically occurs within three days following infection due to secondary pathogenic bacterial invasion [3].

Due to the lack of registered biocontrol agents, farmers use toxic chemicals including formalin for the removal of parasites in aquaculture industries [14]. The chemicals are extremely harmful to fish, humans and the environment [15]. Thus, it is very important to develop a natural control agent to minimize the application of toxic chemicals. A plant can be used as a biocontrol agent due to the presence of different antiparasitic phytochemicals. *Nephrolepis biserrata* (Nephrolepidaceae) is a tropical fern known as “Paku larat” in Malay [16,17]. The plant is a good source of antioxidant compounds with hepatoprotective potential [18]. The antimicrobial activity of the plant has been reported [19], however, the antiparasitic potential of the plant has not yet been reported. Thus, in the current study, we analyzed the antiparasitic potential of the chromatographic fractions of *N. biserrata* methanol extract against the marine parasitic leech and profiled the phytochemical composition via LC-QTOF-MS analysis.

## 2. Results

### 2.1. Antiparasitic Activity of the Chromatographic Fractions of N. biserrata Methanol Extract

The sample plant was extracted with methanol and further chromatographic fractions of the methanol extract were obtained by flash column chromatography techniques. Various concentrations of the chromatographic fractions of *N. biserrata* methanol extract were prepared and the leeches were exposed (Figure 1). The death time and percentage of the seawater-, formalin- and fractions-treated groups are indicated in Table 1 (Appendix A). No mortality was noticed in the negative control group (Figure 1a) while 100 per cent mortality was recorded in the positive control group treated with 0.25% *v/v* of formalin solution (Figure 1b). In the groups treated with fractions 1, 2, 6 and 7 no mortalities of the leeches were observed while complete mortality was noticed in the groups treated with fractions 3, 4 and 5. In this regard, the group treated with fraction 3 killed all the leeches in a time limit of 1.92 ± 0.59 min at a concentration of 2.50 mg/mL, while fractions 4 and 5 killed all the leeches in 34.57 ± 3.39 (14.0 mg/mL) and 36.82 ± 4.53 min (15.3 mg/mL) (Figure 1c,d). The leeches were considered as dead only when they were not able to move their body even after touching with forceps (Figure 1b–d).

### 2.2. Behavioural Changes Induced by the Chromatographic Fractions of N. biserrata Methanol Extract

The leeches treated with seawater indicated no behavioral changes. They were attached firmly to the bottom of the plate using their caudal or posterior sucker, demonstrating a normal swimming pattern (Figure 1). The exposure of parasitic leeches to fractions 3, 4 and 5 groups resulted in behavioral changes. The parasitic leeches treated with fraction 3 showed aggressive movements for the first 0.5–1 min. Then, the leeches were not able to attach using their posterior suckers to the bottom of the plate. After 1 min., the leeches became too weak, stopped movement and were dead eventually. Similarly, the leeches exposed to fractions 4 and 5 indicated aggressive swimming for the first 5–10 min. and then their activity decreased slowly. After 25–30 min., the leeches became too weak and showed no movement, even after touched using forceps and finally considered them as died. No behavioral changes were noticed in the groups treated with fractions 1, 2, 6 and 7 of the chromatographic fractions of *N. biserrata* methanol extract, they were swimming normally and attached firmly to the plate surface.

### 2.3. Phytochemical Compounds Detected in the Chromatographic Fractions of N. biserrata Methanol Extract

All seven frictions of *N. biserrata* extract were subjected to LC-QTOF-MS analysis. Among the 566 molecular features extracted via LC-QTOF-MS analysis, a total of 34 secondary metabolites and one chlorophyll breakdown compound were matched successfully. Some of these secondary metabolites were pyrethrosin (**1**), ivalin (**2**), warburganal (**3**), isodomedin (**4**), 3′-deoxydryopteric acid (**5**), phytosphingosine (**6**), 8-oxo-9,11-octadecadiynoic acid (**7**), 10,16-heptadecadien-8-ynoic acid (**8**), haplophytine (**9**), 5,2′,4′-trihydroxy-3,7,8,5′-tetramethoxy flavone (**10**), 3,5,6,7-tetramethoxy flavone (**11**), 3,4,5-trimethoxycinnamic acid (**12**), 3,4-methylenedioxy-2′,4′-dimethoxychalcone (**13**), 9-tetradecenyl acetate (**14**), 3,7,17,23-tetrahydroxycholan-24-oic acid (**15**), and pheophorbide a (**16**). In overall, we successfully matched six terpenoids, four phenolics, one sphingoid, four fatty acids, two alkaloids, six flavonoids, six aromatics, one aliphatic acetate and four steroids. The retention time, molecular formula and class of the secondary metabolites matched in the fractions of *N. biserrata* via LC-QTOF-MS analysis are shown in Table 2 along with their chromatograms (Figure 2a–c). While the chemical structures of fraction 3, 4 and 5 metabolites are indicated in Figure 3a–c.

## 3. Discussion

To minimize the consumption of toxic chemicals against parasites in aquaculture industries, development and application of natural agent is crucial [7,20]. For this purpose, the phytotherapy of fish disinfestation against parasites and diseases is a good alternative due to the presence of various phytochemical compounds [21]. The chromatographic fractions 3, 4 and 5 of the methanol extract of *N. biserrata* demonstrated significant antiparasitic potential against the marine parasitic leech. 

Compared to our study some other plants have also been reported with antiparasitic properties such as the chromatographic fractions of the methanol extract of *Dillenia suffruticosa* (Dilleneacea) (known as “Simpoh ayer” in Malay) [22]. A fraction of the methanol extract of *D. suffruticosa* resulted in the total mortality of *Z. arugamensis* at a concentration of 20 mg/mL in a time limit of 31 min [22]. The active component of *Piper nigrum* (Piperaceae) (black pepper) has been applied against branchiuran ectoparasite *Argulus* spp. infesting goldfish *Carassius auratus* at a concentration ranging from 1 to 9 mg/L. Total mortality (100%) of the parasite has been reported at 9.0 mg/L in 3 h [23]. The methanol extract of *Radix Bupleuri chinensis* (roots of *Bupleuri chinensis*, known as “Chai Hu” in Chinese) (Umbelliferae) tested for their in vivo antiparasitic properties against monogenean *Dactylogyrus intermedius* in goldfish. The parasites were exposed to the extract at a concentration of 10 mg/L and more than 90% of elimination was recorded in 48 h [24]. The methanol extract of *Magnolia officinalis* (Magnoliaceae) (known as “Houpo” in Chinese) and *Sophora alopecuroides* (Leguminosae) (known as “Ku dou zi” in Chinese) plants have been reported as having antiparasitic properties [25,26,27]. The methanol extract of *M*. *officinalis* and *S. alopecuroides* at a concentration of 40 mg/L and 120 mg/L, respectively, were applied against the ciliate parasite *Ichthyophthirius multifiliis* infesting goldfish for 1 h. The extract of the plants reduced the prevalence of parasites by 24.7% and 44.7%, respectively [27]. From the abovementioned different studies, it is evidently proved that the medicinal plants have the potent antiparasitic properties [22,23,24,25,26,27]. Compared to these plants, the chromatographic fractions of the methanol extract of our sample have shown much better activity such as fraction 3 killing all leeches in 1.92 min. In addition to this, the mortality time shown by fraction 3 was almost half compared to formalin (3.90 min) (Table 1). It is interesting to reveal that the fraction 3 of *N. biserrata* is more effective than formalin in addition to the safe and healthy nature of the plant [7,15,28].

The LCMS analysis indicated the presence of various metabolites with antimicrobial, anti-inflammatory and antiparasitic potential [29,30,31,32,33,34,35]. Though there is a lack of information regarding the antiparasitic potential of these metabolites against marine parasites, their effectiveness against other parasites were widely reported [29,30,31,32,33,34,35]. Some of the important metabolites are phytosphingosine (**6**), a naturally occurring lipid which acts as antimicrobial (gram-positive bacteria, yeast and moulds) and anti-inflammatory [29]. It also induces mitochondria-mediated apoptosis (cell death) by the fragmentation of chromatin DNA in human T-cell lymphoma Jurkat cells [30]. Pheophorbide a (**16**), a chlorophyll breakdown product, has been reported to have antiparasitic properties against *Leishmania amazonensis*, responsible for the disease leishmaniasis [31]. Additionally, the bioactive compound has been reported as having antiviral, antiinflammatory, antioxidant and immunomodulatory properties [32,33,34,35]. Haplophytine (**9**), an alkaloid, has been reported to have insecticidal properties [36]. These 3 metabolites are detected in fraction 3 of *N. biserrata* which indicated very strong antiparasitic properties by killing all the leeches in less than 2 min (Table 1). It is suggested that the effectiveness of fraction 3 could be due to the presence of these metabolites. Ivalin (**2**), a terpenoid, has been reported to have antiparasitic activity against the *Trypanosoma brucei rhodesiense* and *T. cruzi* [37]. It also induces apoptosis in human hepatocellular carcinoma SMMC-7721 Cells [38]. Another terpenoid, warburganal (**3**), along with polygodial bioactive compounds, has been reported with antihelminthic (*Caenorhabditis elegans*) activity [39]. Warburganal has also been reported with molluscicidal properties [40]. Isodomedin (**4**), a diterpene, has been reported with antifeedant properties against the larvae of African armyworm, *Spodoptera exenzpta* [41]. These compounds have been detected in fractions 4 and 5 of *N. biserrata* (Table 2). Though different types of bioactive compounds were also detected in fractions 1, 2, 6, and 7, as well, no significant inhibition or behavioral changes of *Z. arugamensis* were observed. These phenomena might be due to the resistance of *Z. arugamensis* toward these compounds or the presence of low contents of these bioactive compounds.

## 4. Materials and Methods

### 4.1. Chemicals

A mixture of hexane and ethyl acetate was used for column preparation and fractionation. Silica gel (SiO_2_) and normal phase thin layer chromatography plate were obtained from Merck, Germany. Developed TLCs were visualized using 5 per cent molybdophosphoric acid in ethanol was purchased from Nacalai, Japan. Formalin (37 per cent aqueous formaldehyde solution) was bought from Sigma, Leica, Microsystem (Darmstadt, Germany), Methanol (HPLC grade) was purchased from Merck (Darmstadt, Germany). LCMS-grade acetonitrile was obtained from J. T. Baker (Philipsburg, NJ, USA). Polyvinylidene fluoride (PVDF) syringe filters (0.22 µm pore size and 13 mm diameter) were bought from Merck (Darmstadt, Germany). Deionized water was acquired via a Milli-Q system (Merck, Darmstadt, Germany) at a resistivity of >18.2 MΩ·cm. Reference mass solution containing 5.0 mM of purine and 2.5 mM of Hexakis [1H,1H,3H-tetrafluoropropoxy] phosphazine, was procured from Agilent Technologies (Santa Clara, CA, USA). LCMS-grade formic acid (HCOOC) was acquired from Fisher Scientific (Fair Lawn, NJ, USA). 

### 4.2. Plant Collection

Aerial parts of the plants were obtained from Universiti Malaysia Sabah (5.7346° N, 115.9319° E) in Sept 2019 on a sunny day with a minimum temperature of 24 and a maximum temperature of 35 °C. The plant was identified, a voucher specimen was deposited at the Institute for Tropical Biology and Conservation, Universiti Malaysia Sabah, Kota Kinabalu, Malaysia (Figure 4). 

### 4.3. Extraction and Fractionation

The leaves of the plants were cleaned with distilled water and dried by oven at 37 °C. The dried sample was grounded in a heavy-duty grinder and placed in an airtight jar. Methanol (300 mL, HPLC grade) was used for the extraction of the powder (60 g) via the Soxhlet method (50–60 °C for 72 h). The dry extract was recovered using a vacuum rotary evaporator, before lyophilization using a freeze drier [42].

Using a 20 × 100 mm glass column with a 250 mL reservoir, flash column chromatography was carried out. The column was prepared with SiO_2_ gel infused in hexane. Around 500 mg of the methanol extract of *N. biserrata* was dissolved in 2 mL of hexane and spiked onto the column once the column settled. The spiked silica gel (SiO2) was eluted with a solvent-gradient Hexane-Ethyl Acetate eluant system; F1-Hex:EtOAc (9:1), F2-Hex:EtOAc (8:2), F3-Hex:EtOAc (7:3), F4-Hex:EtOAc (6:4), F5-Hex:EtOAc (5:5), F6-Hex:EtOAc (4:6), F7-EtOAc (100%) was used to fractionate. A total of seven fractions were collected, each was about 250 mL in volume, fractions were evaporated under vacuo using a vacuum rotary evaporator, before removing the solvent residues using a N_2_ flushed desiccation. Then, their profiles were determined using thin-layer chromatography to notice their separation. Besides, fractions were subjected to LC-QTOF-MS analysis to determine the inherently present secondary metabolites.

### 4.4. Antiparasitic Bioassay 

The parasitic leeches were isolated from the infested hybrid groupers (*Epinephelus fuscoguttatus* × *E. lanceolatus*), Universiti Malaysia Sabah and identified based on its morphological features [7]. The mature leeches were divided into 9 groups (5 leeches per group) and as follows.

Group 1 = negative control, treated with seawater.

Group 2 = positive control, treated with 0.25% formalin solution.

Group 3 = treated with *N. biserrata* fraction 1 (4.80 mg/mL).

Group 4 = treated with *N. biserrata* fraction 2 (20 mg/mL).

Group 5 = treated with *N. biserrata* fraction 3 (2.50 mg/mL).

Group 6 = treated with *N. biserrata* fraction 4 (14 mg/mL).

Group 7 = treated with *N. biserrata* fraction 5 (15.30 mg/mL).

Group 8 = treated with *N. biserrata* fraction 6 (17.20 mg/mL).

Group 9 = treated with *N. biserrata* fraction 7 (20 mg/mL).

Different concentration of fractions was applied according to the fractionation yield ratio of the methanol extract of *N. biserrata* to identify the effective fractions against *Z. arugamensis*. The time limit was recorded until all the leeches were killed after the exposure to various concentrations of chromatographic fractions of methanol extract of *N. biserrata* [43].

### 4.5. Behavioural Observation

The behavior of *Z. arugamensis* was also monitored visually during the exposure to different concentrations of the chromatographic fractions of *N. biserrata* and compared with normal and positive control groups [43].

### 4.6. LC-QTOF MS Acquisition

The fractions were analyzed as described previously [22], where Agilent 1290 Infinity LC system coupled with an Agilent 6520 QTOF-MS system was employed. Briefly, a 2.0 µL sample was injected and separated with an Agilent Zorbax Eclipse XDB-C18 column (narrow bore, 2.1 mm × 150 mm × 3.5 µm; Agilent Technologies, Santa Clara, CA, USA) at a flow rate of 0.5 mL/min. Column chamber was maintained at 25 °C during analysis. Solvents A (H_2_O-0.1% HCOOH) and B (acetonitrile-0.1% HCOOH) were used as the mobile phases, and the gradient was programmed at 5% solvent B from 0 to 5 min, then from 5% to 100% solvent B in 15 min and kept for 5 min, before column re-conditioning as initial for next injection. 

Mass spectrometry (MS) data acquisition range was set from *m/z* of 100 to 1500. Heated electrospray ionization (ESI) was deployed at 4 kV in positive mode. The ion source was set as follows: 300 °C of gas temperature, 10 L/min of drying gas flow, and 45 psig of nebulizer flow. Tuning Mix (Agilent Technologies, Santa Clara, CA, USA) was used for system calibration before analysis. During analysis, internal mass calibration standards betaine and hexakis (1H, 1H, 3H-tetrafluoropropoxy) phosphazine were introduced. In positive mode, the internal mass calibration standards were *m/z* of 121.0508 and 922.0097, respectively. 

Automated tandem mass spectrometry (auto-MS/MS) was performed for metabolites matching, on which two precursor ions were selected from each MS scan and subjected for collision-induced dissociation with 20 eV of collision energy. Data acquisition range for auto-MS/MS was set between an *m/z* of 50 and 1500, and purified nitrogen gas was used as the collision gas. Acquired data were processed using Agilent MassHunter Qualitative Workflows software (version B.08.00) and the extracted metabolite MS were identified via METLIN metabolites and lipids databases [44] with 2 ppm maximum mass tolerance. 

### 4.7. Statistical Analysis

Data analysis was done via IBM SPSS Statistics 25 Window package (IBM, Armonk, NY, US). One-way variance analysis (ANOVA) followed by Tukey’s multiple comparison test was applied to determine the significant differences between groups All findings were presented as mean ± standard deviation (S.D.). *p*-Values of less than 0.05 were considered to be significant.

## 5. Conclusions

In this study, fractions 3, 4 and 5 of *N. biserrata* demonstrated antiparasitic properties among the seven fractions obtained through flash column chromatography techniques. Significant parasitic mortality was indicated by fraction 3 at a concentration of 2.5 mg/mL, all the parasitic leeches were killed in a time limit of only 1.92 ± 0.59 min. followed by fraction 4 (14 mg/mL) 34.57 ± 3.39 and fraction 5 (15.3 mg/mL) in an average time of 36.82 ± 4.53 min. In a very low concentration (2.5 mg/mL), the methanol extract of *N. biserrata* could effectively kill the parasites infesting the cultured grouper. LC-QTOF-MS analysis indicated the presence of secondary metabolites including phytosphingosine (**6**), pyrethrosin (**1**), haplophytine (**9**), ivalin (**2**), warburganal (**3**), isodomedin (**4**) and pheophorbide a (**16**) representing sphingoid, alkaloid, terpenoid, phenolic and flavonoid groups. Thus, our study indicated that *N. biserrata* fractions contained some effective and potent bioactive compounds with antiparasitic potential.

## Figures and Tables

**Figure 1 molecules-26-00499-f001:**
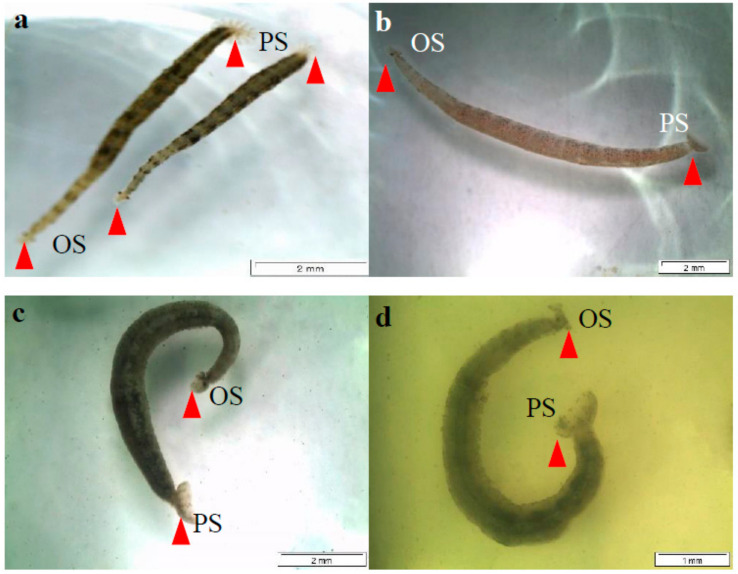
*Zeylanicobdella arugamensis*, arrow indicates both OS = oral or anterior sucker and PS = posterior or caudal sucker. (**a**) = treated with seawater only, arrow indicated to show the posterior sucker attached firmly to the bottom of the plate, (**b**) = treated with chemical formalin, here posterior sucker detached, (**c**,**d**) = treated with chromatographic fractions of the methanol extract of *N. biserrata*, showing the detachment of posterior sucker.

**Figure 2 molecules-26-00499-f002:**
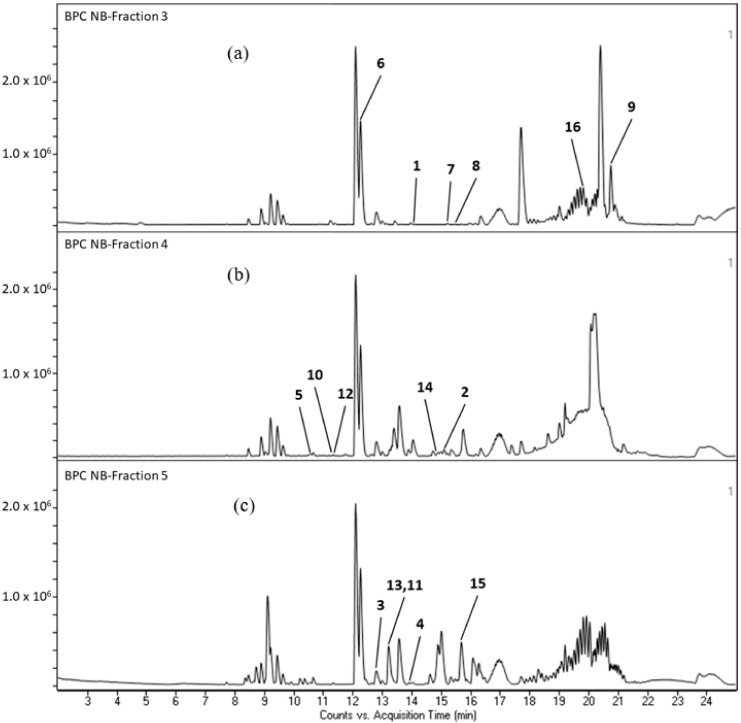
(**a**) Chromatogram of fraction 3 of *N. biserrata*. Pyrethrosin (**1**), phytosphingosine (**6**), 8-oxo-9,11-octadecadiynoic acid (**7**), 10,16-heptadecadien-8-ynoic acid (**8**), haplophytine (**9**), pheophorbide a (**16**). (**b**) Chromatogram of fraction 4. Ivalin (**2**), 3′-deoxydryopteric acid (**5**), 5,2′,4′-trihydroxy-3,7,8,5′-tetramethoxyflavone (**10**), 3,4,5-trimethoxycinnamic acid (**12**), 9-tetradecenyl acetate (**14**). (**c**) Chromatogram of fraction 5. Warburganal (**3**), isodomedin (**4**), 3,5,6,7-tetramethoxyflavone (**11**), 3,4-methylenedioxy-2′,4′-dimethoxychalcone (**13**), 3,7,17,23-tetrahydroxycholan-24-oic acid (**15**).

**Figure 3 molecules-26-00499-f003:**
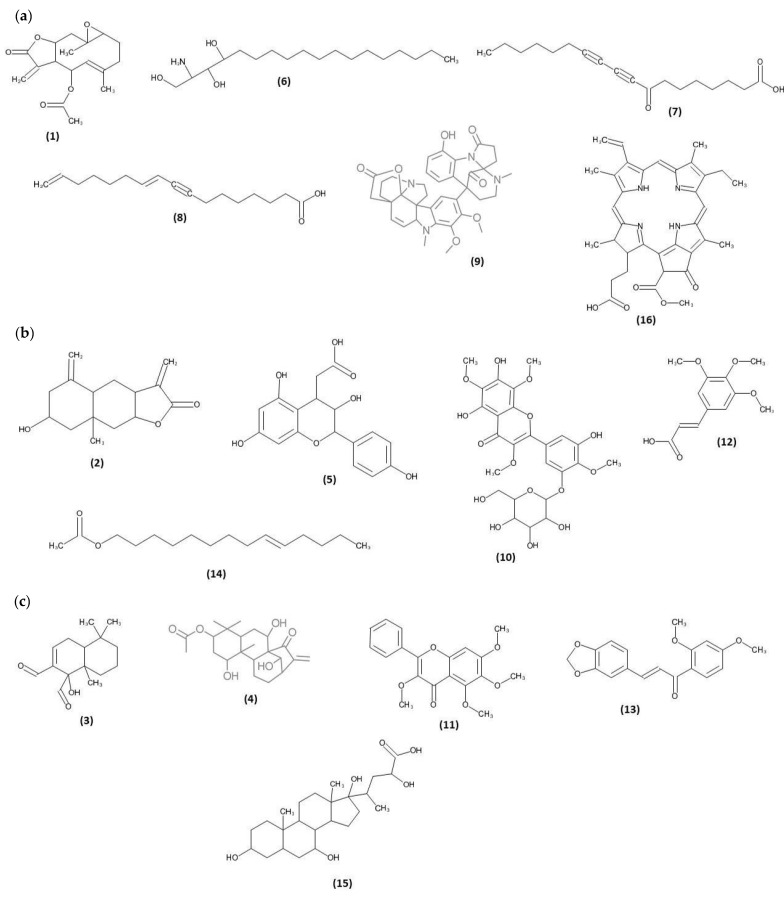
(**a**) Chemical structures of the secondary metabolites detected in the chromatographic fraction 3 of *N. biserrata*. Pyrethrosin (**1**), phytosphingosine (**6**), 8-oxo-9,11-octadecadiynoic acid (**7**), 10,16-heptadecadien-8-ynoic acid (**8**), haplophytine (**9**), pheophorbide a (**16**). (**b**) Chemical structures of the secondary metabolites detected in the chromatographic fraction 4. Ivalin (**2**), 3′-deoxydryopteric acid (**5**), 5,2′,4′-trihydroxy-3,7,8,5′-tetramethoxyflavone (**10**), 3,4,5-trimethoxycinnamic acid (**12**), 9-tetradecenyl acetate (**14**). (**c**) Chemical structures of the secondary metabolites detected in the chromatographic fraction 5. Warburganal (**3**), isodomedin (**4**), 3,5,6,7-tetramethoxyflavone (**11**), 3,4-methylenedioxy-2′,4′-dimethoxychalcone (**13**), 3,7,17,23-tetrahydroxycholan-24-oic acid (**15**).

**Figure 4 molecules-26-00499-f004:**
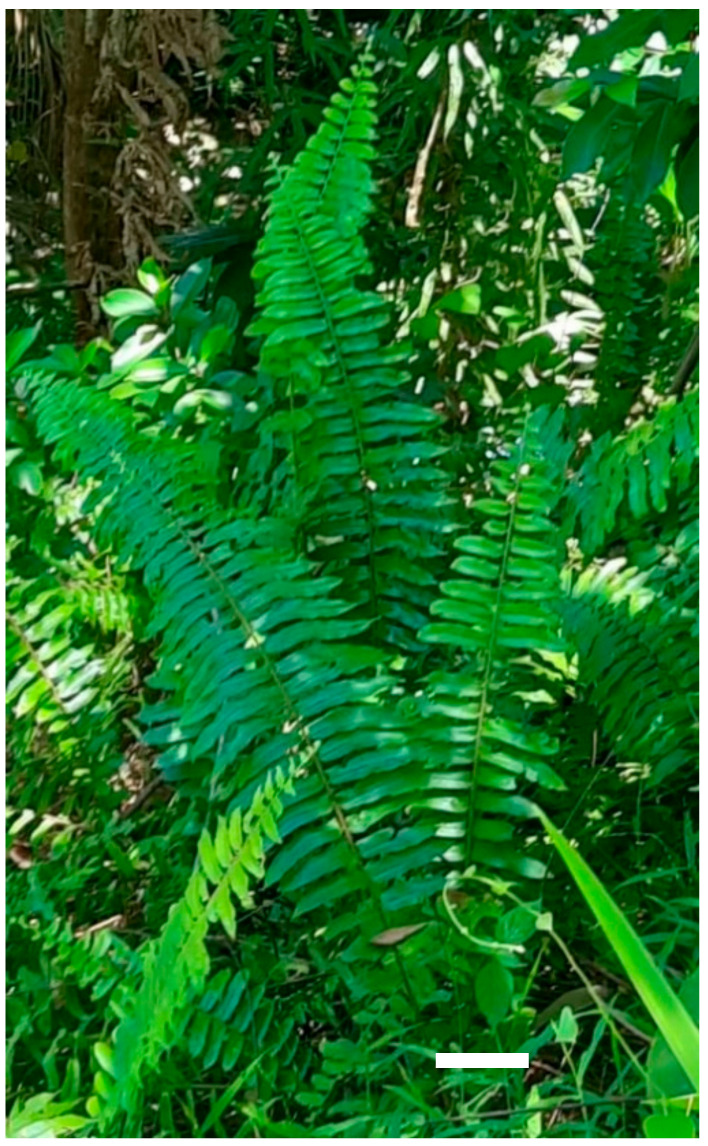
*Nephrolepis biserrata* fern collected from Universiti Malaysia Sabah, scale bar: 5 cm.

**Table 1 molecules-26-00499-t001:** Death time and percentage of the fractions of *N. biserrata* against the marine leeches.

No	Group	Death Time (min) Mean ± S.D.	Death Percentage
1	Negative Control	0 00 ± 0.00	0
2	Positive Control (Formalin 0.025%)	3.90 ± 0.84 ^#^	100
3	*Fraction 1* (4.80 mg/mL)	0 00 ± 0.00 *	0
4	*Fraction 2* (20 mg/mL)	0 00 ± 0.00 *	0
5	*Fraction 3* (2.50 mg/mL)	1.92 ± 0.59 ^#^ * ^^ $^	100
6	*Fraction 4* (14 mg/mL)	34.57 ± 3.39 ^#^ * ^^ $ &^	100
7	*Fraction 5* (15.30 mg/mL)	36.82 ± 4.53 ^#^ * ^^ $ &^	100
8	*Fraction 6* (17.20 mg/mL)	0 00 ± 0.00 * ^& α ®^	00
9	*Fraction 7* (20 mg/mL)	0 00 ± 0.00 * ^& β ®^	00

Values are mean ± SD of six leeches per group. *Fraction 1* [hexane: ethyl acetate (H:E) 9:1)]; *Fraction 2* (H:E, 8:2); *Fraction 3* (H:E, 7:3); *Fraction 4* (H:E, 6:4); *Fraction 5* (H:E, 5:5); *Fraction 6* (H:E, 4:6), *Fraction* 7 (E, 100%). ^#^ Significance at *p* < 0.05 compared with the negative control group. * Significance at *p* < 0.05 compared with the formalin [0.25% (*v*/*v*)]. ^ Significance at *p* < 0.05 compared with fraction 1 (4.80 mg/mL). ^$^ Significance at *p* < 0.05 compared with fraction 2 (20 mg/mL). ^&^ Significance at *p* < 0.05 compared with fraction 3 (2.50 mg/mL). ^α^ Significance at *p* < 0.05 compared with fraction 4 (14 mg/mL). ^®^ Significance at *p* < 0.05 compared with fraction 5 (15.30 mg/mL). ^β^ Significance at *p* < 0.05 compared with fraction 6 (17.20 mg/mL).

**Table 2 molecules-26-00499-t002:** Secondary metabolite profiles in fractions of *N. biserrata* analyzed via LC-QTOF-MS.

Fractions	Retention Time (RT)	Mass to Charge Ratio, *m/z*	Formula	Mass Error, ppm	Matched Metabolites	Class
1	13.144	203.1043	C_11_ H_16_ O_2_	−1.39	3-tert-Butyl-5-methylcatechol	Phenolic
2	12.848	231.1377	C_15_ H_18_ O_2_	0.30	Isodehydrocostus lactone	Terpenoid
	13.112	203.1045	C_11_ H_16_ O_2_	−1.39	3-tert-Butyl-5-methylcatechol	Phenolic
	13.443	311.1847	C_17_ H_26_ O_5_	−0.46	methyl 8-[2-(2-formyl-vinyl)-3-hydroxy-5-oxo-cyclopentyl]-octanoate	Fatty acid methyl ester
	16.533	280.1905	C_16_ H_25_ N O_3_	−0.37	Serratine	Alkaloid
	19.234	425.2905	C_24_ H_40_ O_6_	−1.51	(3α,5β,6α,7α)-2,3,6,7-Tetrahydroxycholan-24-oic acid	Steroid
	24.443	431.3522	C_28_ H_46_ O_3_	−0.03	1,25-Dihydroxy-ethylidene-19-norvitamin D_3_	Steroid
3	12.220	318.3001	C_18_ H_39_ N O_3_	0.93	Phytosphingosine (**6**)	Sphingoid
	14.096	307.1541	C_17_ H_22_ O_5_	−0.38	Pyrethrosin (**1**)	Terpenoid
	15.081	291.1954	C_18_ H_26_ O_3_	−0.04	8-oxo-9,11-octadecadiynoic acid (**7**)	Oxo Fatty acid
	15.343	263.2002	C_17_ H_26_ O_2_	1.19	10,16-Heptadecadien-8-ynoic acid (**8**)	Fatty Acid
	19.835	593.2766	C_35_ H_36_ N_4_ O_5_	−0.86	Pheophorbide a (**16**)	Chlorophyll breakdown product
	20.777	653.2969	C_37_ H_40_ N_4_ O_7_	0.46	Haplophytine (**9**)	Alkaloid
4	10.663	333.0965	C_17_ H_16_ O_7_	1.05	3′-Deoxydryopteric acid (**5**)	Phenolic
	11.325	391.1018	C_19_ H_18_ O_9_	1.34	5,2′,4′-Trihydroxy-3,7,8,5′-tetramethoxyflavone (**10**)	Flavonoid
	11.404	239.0911	C_12_ H_14_ O_5_	0.11	3,4,5-Trimethoxycinnamic acid (**12**)	Aromatic
	14.927	277.2138	C_16_ H_30_ O_2_	0.12	9-Tetradecenyl acetate (**14**)	Aliphatic acetate
	15.120	249.1483	C_15_ H_20_ O_3_	−0.32	Ivalin (**2**)	Terpenoid
5	12.796	251.1636	C_15_ H_22_ O_3_	1.86	Warburganal (**3**)	Terpenoid
	13.223	313.1070	C_18_ H_16_ O_5_	0.75	3,4-Methylenedioxy-2′,4′-dimethoxychalcone (**13**)	Aromatic
	13.245	343.1176	C_19_ H_18_ O_6_	0.27	3,5,6,7-Tetramethoxyflavone (**11**)	Flavonoid
	13.964	415.2085	C_22_ H_32_ O_6_	1.05	Isodomedin (**4**)	Terpenoid
	15.651	447.2719	C_24_ H_40_ O_6_	−1.27	3,7,17,23-Tetrahydroxycholan-24-oic acid (**15**)	Steroid
6	8.965	465.1018	C_21_ H_20_ O_12_	1.83	Herbacetin 3-glucoside	Flavonoid
	10.052	611.1391	C_30_ H_26_ O_14_	1.04	Gallocatechin-(4alpha- > 8)-epigallocatechin	Flavonoid
	12.306	290.269	C_16_ H_32_ O_3_	0.26	14-hydroxy-hexadecanoic acid	Fatty acid
	13.230	343.117	C_19_ H_18_ O_6_	1.78	3,5,6,7-Tetramethoxyflavone	Flavonoid
	13.983	415.2086	C_22_ H_32_ O_6_	1.55	Isodomedin	Terpenoid
	21.127	391.2845	C_24_ H_38_ O_4_	−0.89	3,7-Dihydroxychol-22-en-24-oic acid	Steroid
7	2.053	328.1393	C_15_ H_18_ O_7_	−0.86	Picrotin	Terpenoid
	7.065	163.0386	C_9_ H_6_ O_3_	1.15	3-Hydroxycoumarin	Aromatic
	7.309	355.1016	C_16_ H_18_ O_9_	1.62	Chlorogenic Acid	Phenolic
	8.945	465.1020	C_21_ H_20_ O_12_	1.89	Herbacetin 3-glucoside	Flavonoid
	9.008	291.0975	C_14_ H_14_ N_2_ O_5_	−0.40	N2-Malonyl-D-tryptophan	Aromatic
	9.333	303.0500	C_15_ H_10_ O_7_	−1.53	Melanoxetin	Flavonoid
	9.333	449.1077	C_21_ H_20_ O_11_	0.25	Kaempferol 5-glucoside	Flavonoid
	10.034	404.1341	C_20_ H_18_ O_8_	−0.71	Glucosyloxyanthraquinone	Aromatic
	12.296	290.2687	C_16_ H_32_ O_3_	1.23	14-hydroxy-hexadecanoic acid	Fatty acid
	13.994	432.2381	C_24_ H_30_ O_6_	−0.39	Magnoshinin	Aromatic
	17.923	506.2541	C_30_ H_32_ O_6_	−1.45	Rubraflavone D	Phenolic

## Data Availability

The data presented in this study are available in manuscript and Appendix A.

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
