# Peer review of "Antiparasitic Potential of Chromatographic Fractions of Nephrolepis biserrata and Liquid Chromatography-Quadrupole Time-of-Flight-Mass Spectrometry Analysis"

_molecules, 2021, doi:10.3390/molecules26020499_

Round 1

Reviewer 1 Report

This is a well-written paper that describes potential anti-leech metabolites produced by a tropical fern, Nephrolepis biserrata.   The results presented are interesting and promising.  However, the work as presented appears to be un-replicated, and further analytical information is needed.

Specific concerns:

  1. How many times was the experiment presented in Table 1 replicated? The data appear to be from a single experiment with a total of 6 leeches per treatment. It is important to show that these results are reproducible.  Even if all fractions are not re-tested, the relevant fractions should be retested with larger numbers of leeches.  The experiment presented in section 2.2 further shows an effect of these fractions, but it is not clear that this was done with a different group of leeches than the data presented in Table 1.  Also, even if different leeches, was it done with the same set of extracts (i.e., not a true biological replication)?
  2. mg/ml is an extremely high concentration. Was it mg/ml or mg/L? mg/L would be consistent with the cited references (e.g., 23, 24).
  3. how were mg of extract determined? Were the samples dried? If not, why use mg instead of ml?  The methods suggest they were concentrated, but not dried.  If concentrated, what was the extent of concentration?  How was that determined?  If the extent of concentration was not consistent among fractions, how were comparisons made among different fractions?
  4. were we to think that fraction 3 was 6-fold less concentrated than factions 4 or 5? If so, and given that the death time was much 17x faster, it seems this would suggest that the key component was nearly exclusively in fraction 3. If this is the case, which compounds were uniquely abundant in fraction 3?
  5. Additional information needs to be presented on how the compounds discussed in section 2.3/Table 2 were identified. The methods indicated that two precursor ions were selected from each MS scan for fragmentation, but 5-6 molecules were listed for each fraction in Table2.

Other items:

  1. Since methods appear after results, further information needs to be provided in the results section. Section 2.1 should provide a general description of the source of the extracts, how they were prepared, how the leeches were determined to be dead.
  2. the definition of the fractions (e.g., H:E 9:1) should be directly in the Table rather than in the footnote.
  3. Fig 2 does not need to show the fractions that were not biologically relevant; those can be in supplemental data.
  4. Fig 3 should indicate which molecules were in which fraction and should be arranged within the figure according to fraction. Alternatively, maybe only show those from fraction 3.
  5. line 165-166. Toxic chemicals, even if produced by a plant, are not safe and healthy.

Author Response

R1- Responses to the editor and reviewer comments

Ref: Manuscript ID: molecules-1038495

Title: Antiparasitic Potential of Chromatographic Fractions of Nephrolepis
biserrata and Liquid Chromatography–Quadrupole Time-of-Flight–Mass
Spectrometry Analysis

Dear Editor,

Thank you for your valuable comments and suggestions which are very useful to improve our manuscript.

We have given our responses to all the queries raised by the reviewers.

REVIEWER 1 COMMENTS

This is a well-written paper that describes potential anti-leech metabolites produced by a tropical fern, Nephrolepis biserrata.   The results presented are interesting and promising.  However, the work as presented appears to be un-replicated, and further analytical information is needed.

Specific concerns:

Point 1:  How many times was the experiment presented in Table 1 replicated? The data appear to be from a single experiment with a total of 6 leeches per treatment. It is important to show that these results are reproducible.  Even if all fractions are not re-tested, the relevant fractions should be retested with larger numbers of leeches.  The experiment presented in section 2.2 further shows an effect of these fractions, but it is not clear that this was done with a different group of leeches than the data presented in Table 1.  Also, even if different leeches, was it done with the same set of extracts (i.e., not a true biological replication)?

Res: Thank you for your comments.

The experiment was performed in triplicate. In each experiment, 6 leeches per group were used. For every individual experiment, the related concentration of freshly prepared fractions was used and a new batch of leeches was taken from different infested fishes from my research locality within Universiti Malaysia Sabah Fish Hatchery.

Point 2: mg/ml is an extremely high concentration. Was it mg/ml or mg/L? mg/L would be consistent with the cited references (e.g., 23, 24).

Res: Thank you for your comments.

The main objective of our study was to highlight the antiparasitic potential of the chromatographic fractions of Nephrolepis biserrata and we reported the effective fractions particularly fraction 3 (2.5 mg/ml). As far as the concentration is concerned we selected the maximum concentrations of the fractions for our study to know about their effectiveness. The applied concentrations against the leeches can be diluted but it will also increase the mortality time of the leeches. In future studies, we will also reduce the concentration of the fractions against the infested fishes.

Point 3: how were mg of extract determined? Were the samples dried? If not, why use mg instead of ml?  The methods suggest they were concentrated, but not dried.  If concentrated, what was the extent of concentration?  How was that determined?  If the extent of concentration was not consistent among fractions, how were comparisons made among different fractions?

Res: Thank you for your comments.

The sample was extracted with 100% HPLC grade Methanol. The dry extract was recovered using a vacuum rotary evaporator, before lyophilization using a freeze drier. The methanol extract of N. biserrata was dissolved in 2 ml of hexane and spiked onto the column once the column settled. The spiked silica gel (SiO2) was eluted with a solvent-gradient Hexane-Ethyl Acetate eluant system; F1 - Hex:EtOAc (9:1), F2 - Hex:EtOAc (8:2), F3 - Hex:EtOAc (7:3), F4 - Hex:EtOAc (6:4), F5 - Hex:EtOAc (5:5), F6 - Hex:EtOAc (4:6), F7 – EtOAc (100 %) was used to fractionate.  A total of seven fractions were collected, each was about 250 mL in volume, fractions were evaporated under vacuo using a vacuum rotary evaporator, before removing the solvent residues using a N2 flushed desiccation. In the end, dry fractions were obtained and thus the concentrations were determined in mg.

Point 4: were we to think that fraction 3 was 6-fold less concentrated than factions 4 or 5? If so, and given that the death time was much 17x faster, it seems this would suggest that the key component was nearly exclusively in fraction 3. If this is the case, which compounds were uniquely abundant in fraction 3?

Res: Thank you for your comments.

Yes, fraction 3 is the most effective fractions among all, the LC-QTOF-MS indicated the presence of 3 antiparasitic bioactive compounds. Further research work will be carried out to purify fraction 3 and to prove whether the present compounds will be effective in the pure stage or they will show effectiveness only in the mixture form due to the synergistic effect.

Point 5: Additional information needs to be presented on how the compounds discussed in section 2.3/Table 2 were identified. The methods indicated that two precursor ions were selected from each MS scan for fragmentation, but 5-6 molecules were listed for each fraction in Table2.

Res: Thank you for your comments.

In MS analysis, there is impossible to have all metabolites identified or matched. In current LC-QTOF analysis, data acquisition rate for MS is 3 spectra/second while MSMS is 4 spectra/second. The scanning protocol for auto-MSMS analysis is acquired each MS scan will be followed by 2 MSMS scan, as two precursor ions will be selected from each MS scan for fragmentation. Then, several MSMS spectra (at least 3) from the same precursor ion will be average and yield a final MSMS spectra (metabolite feature) for metabolite matching against METLIN database. In each LC-QTOF analysis, we would be able to acquire about 300 metabolite features for database matching. After background removal, we left about 100 metabolite features. Lastly, considering the limitation of database and instrument’s resolution, most of the metabolite features are unidentified. Thus, only 5-6 metabolites are matched with high confident level.

Other items:

Point 6: Since methods appear after results, further information needs to be provided in the results section. Section 2.1 should provide a general description of the source of the extracts, how they were prepared, how the leeches were determined to be dead.

Res: Thank you for the comment,

The information has been added in section 2.1 as

“The sample plant was extracted with methanol and further chromatographic fractions of the methanol were obtained by flash column chromatography techniques. Various concentration of the chromatographic fractions of N. biserrata methanol extract was prepared and the leeches were exposed.

The leeches were considered dead once they were not able to show any movement even by physical touch.

Point 7: The definition of the fractions (e.g., H:E 9:1) should be directly in the Table rather than in the footnote.

Res: Thank you for the comment,

The changes have been made in the text as

“The spiked silica gel (SiO2) was eluted with a solvent-gradient Hexane-Ethyl Acetate eluant system; F1 - Hex:EtOAc (9:1), F2 - Hex:EtOAc (8:2), F3 - Hex:EtOAc (7:3), F4 - Hex:EtOAc (6:4), F5 - Hex:EtOAc (5:5), F6 - Hex:EtOAc (4:6), F7 – EtOAc (100 %) was used to fractionate. A total of seven fractions were collected, each was about 250 mL in volume, fractions were evaporated under vacuo using a vacuum rotary evaporator, before removing the solvent residues using a N2 flushed desiccation”.

Point 8: Fig 2 does not need to show the fractions that were not biologically relevant; those can be in supplemental data.

Res: Thank you for the comment,

In Fig 2 the fractions that were not biologically relevant are deleted.

Point 9: Fig 3 should indicate which molecules were in which fraction and should be arranged within the figure according to a fraction. Alternatively, maybe only show those from fraction 3.

Res: Thank you for the comment,

In figure 3 the related molecules in each fraction are arranged and shown in the main text as.

Figure 3a. Chemical structures of the secondary metabolites detected in the chromatographic fractions 3. Pyrethrosin (1), phytosphingosine (6), 8-oxo-9,11-octadecadiynoic acid (7), 10,16-heptadecadien-8-ynoic acid (8), haplophytine (9), pheophorbide a (16).

Figure 3b. Chemical structures of the secondary metabolites detected in the chromatographic fractions 4. Ivalin (2), 3'-deoxydryopteric acid (5), 5,2',4'-trihydroxy-3,7,8,5'-tetramethoxyflavone (10), 3,4,5-trimethoxycinnamic acid (12), 9-tetradecenyl acetate (14),

Figure 3c. Chemical structures of the secondary metabolites detected in the chromatographic fractions 5. Warburganal (3), isodomedin (4), 3,5,6,7-tetramethoxyflavone (11), 3,4-methylenedioxy-2',4'-dimethoxychalcone (13), 3,7,17,23-tetrahydroxycholan-24-oic acid (15)

Point 10: line 165-166. Toxic chemicals, even if produced by a plant, are not safe and healthy.

Res: Thank you for the comment,

Yes, some of the chemicals produced by the plants are also showing toxic to human. In our sample, no toxic compounds have been indicated by LC.QTOF- MS analysis and in addition to this we have previously reported regarding the safe nature of N. biserrata.  The reference for the study is given below.

  1. Shah, M. D., Gnanaraj, C., Haque, A. E. & Iqbal, M. Antioxidative and chemopreventive effects of Nephrolepis biserrata against carbon tetrachloride (CCl4)-induced oxidative stress and hepatic dysfunction in rats. Pharm. Biol. 53, 31–39 (2015).

Reviewer 2 Report

Manuscript Number: molecules-1038495

entitled: Antiparasitic Potential of Chromatographic Fractions of Nephrolepis biserrata and Liquid Chromatography–Quadrupole Time-of-Flight–Mass Spectrometry Analysis

This is a well conducted scientific study, done in a thorough manner and expressed concisely. Therefore, the manuscript is suitable for Molecules after considering the below comments:

  1. Please add or marked chemical structures of the secondary metabolites (1-16) on each Figure 2.
  2. Chemical structures of the secondary metabolites (1-16) drawing on Figure 3 should be re-draw more precisely showing possibly regioisomers (e.g. E/Z or cis/trans).
  3. From the present version of manuscript we do not know which products (the secondary metabolites (1-16)) are main, and which are minor molecules.
  4. Line 268 page 11 there is an information “the extracted metabolite MS were identified via METLIN metabolites and lipids databases [42] with 2 268 ppm maximum mass tolerance.” Did Authors compare your results (the secondary metabolites 1-16) with authentic samples, used as a reference standard?
  5. In my opinion Authors should add information about biological activity of mentioned above the secondary metabolites (1-16).
  6. Please add space between e.g. 300space°C

Author Response

Revision - Responses to the editor and reviewer comments

Ref: Manuscript ID: molecules-1038495

Title: Antiparasitic Potential of Chromatographic Fractions of Nephrolepis
biserrata and Liquid Chromatography–Quadrupole Time-of-Flight–Mass
Spectrometry Analysis

REVIEWER 2 COMMENTS

Thank you for your valuable comments and suggestions which are very useful to improve our manuscript.

This is a well-conducted scientific study, done thoroughly and expressed concisely. Therefore, the manuscript is suitable for Molecules after considering the below comments:

Point 1: Please add or marked chemical structures of the secondary metabolites (1-16) on each Figure 2.

Res: Thank you for the comment,

The marked chemical structures of the secondary metabolites (1-16) on Figure 2 are added in Text as

Figure 2a. Chromatogram of fraction 3 of N. biserrata. Pyrethrosin (1), phytosphingosine (6), 8-oxo-9,11-octadecadiynoic acid (7), 10,16-heptadecadien-8-ynoic acid (8), haplophytine (9), pheophorbide a (16).

Figure 2b. Chromatogram of fraction 4 of N. biserrata. Ivalin (2), 3'-deoxydryopteric acid (5), 5,2',4'-trihydroxy-3,7,8,5'-tetramethoxyflavone (10), 3,4,5-trimethoxycinnamic acid (12), 9-tetradecenyl acetate (14),

Figure 2c. Chromatogram of fraction 5 of N. biserrata. Warburganal (3), isodomedin (4), 3,5,6,7-tetramethoxyflavone (11), 3,4-methylenedioxy-2',4'-dimethoxychalcone (13), 3,7,17,23-tetrahydroxycholan-24-oic acid (15)

Point 2: Chemical structures of the secondary metabolites (1-16) drawing on Figure 3 should be re-draw more precisely showing possibly regioisomers (e.g. E/Z or cis/trans).

Res: Thank you for the comment,

In mass spectrometry analysis, regioisomerism properties are not available. Adding this information into the chemical structure is misleading and might contribute to an erroneous result.

Point 3: From the present version of the manuscript we do not know which products (the secondary metabolites (1-16)) are main, and which are minor molecules.

Res: Thank you for the comment,

Absolute and relative quantifications are not carried out for the current study, and QTOF is not suitable for such determination as well. QTOF is a high-resolution mass spectrometry system with very low sensitivity, thus not suitable for absolute or relative quantification study. Besides, the ionization tendency for each compound is unique and not comparable to each other for relative quantification. Thus, the compound with low MS signals could be the major molecule and vice versa.

Point 4: Line 268 page 11 there is an information “the extracted metabolite MS were identified via METLIN metabolites and lipids databases [42] with 2 268 ppm maximum mass tolerance.” Did Authors compare your results (the secondary metabolites 1-16) with authentic samples, used as a reference standard?

Res: Thank you for the comment

In the current study, untargeted LC-QTOF analysis was carried out without any chemical standard for comparison. All the acquired MSMS data were matched with the METLIN database. In high-resolution mass spectrometry analysis, 5 ppm maximum mass tolerance is used in most published articles, but we aim for higher confident level. Thus, we lower down to 2 ppm maximum mass tolerance, which have higher accuracy.

Point 5: In my opinion Authors should add information about the biological activity of mentioned above the secondary metabolites (1-16).

Res: Thank you for the comment

Further information regarding the biological activities of the metabolites are added in the discussion As.

Line 20-212 : It also induces apoptosis in Human Hepatocellular Carcinoma SMMC-7721 Cells [38].

Warburganal has also been reported with molluscicidal properties [40] Line 212.

Point 6: Please add space between e.g. 300 space°C

Res: Thank you for the comment.

Space is provided.

Round 2

Reviewer 1 Report

1.The most critical objection was lack of replication.  From the additional information provided, it seems that the experiments were replicated.  If so, this information needs to be specifically included in the revised manuscript; both that such experiments were performed, and provide the results of the replicating experiments either in the text or as supplemental data.  

2.The question posed in item 4, 'which compounds were uniquely abundant in fraction 3?', was not answered.  Even if such compounds have not been specifically identified, are there specific compounds/peaks associated with this fraction?

3. Point 5. The difficulty in identifying compounds is why this question was asked.  The additional information about identification provided, including number of peaks and how many were able to identified should be included in the manuscript.  More importantly, though, the fact that the great majority cannot be identified, raises the bigger question, why tell us about the few that can be?  Just because they can be identified, does not make them especially relevant.  Are there biological reasons to tell us about these specific compounds, if not, don't include.

Author Response

Thank you for your valuable comments and suggestions which are very useful to improve our manuscript. Both reviewers have done a wonderful job in their questions and suggestions, reviewer #1 suggestions we followed here and reviewer #2 accepted as such. We have given our responses to all the queries raised by the reviewer #1.

Reviewer #1

Q1.The most critical objection was lack of replication.  From the additional information provided, it seems that the experiments were replicated.  If so, this information needs to be specifically included in the revised manuscript; both that such experiments were performed and provide the results of the replicating experiments either in the text or as supplemental data.  

Res: Thank you very much for the comments

The experiment was performed in triplicate and the result of the experiments are attached as supplementary data.

Q2.The question posed in item 4, 'which compounds were uniquely abundant in fraction 3?', was not answered.  Even if such compounds have not been specifically identified, are there specific compounds/peaks associated with this fraction?

Res: Thank you very much for the comments

The question was answered previously on which the abundancy is unable to be determined via high-resolution mass spectrometry approach. Besides the sensitivity of the mass analyzer, the ionization tendency has to be considered as well. Thus, high-intensity peak(s) are not always correlated to high abundancy. Meanwhile, each peak is consisting MS signals from different ions, and we only selected two major precursor ions from a single MS scan at each time, on which the total intensity of that 2 precursor ions are actually ~5% of the total intensity of MS scan. If insist to label these precursors as major compounds to represent the fraction, it definitely going to mislead the reader.

Q3. Point 5: The difficulty in identifying compounds is why this question was asked.  The additional information about identification provided, including number of peaks and how many were able to identified should be included in the manuscript.  More importantly, though, the fact that the great majority cannot be identified, raises the bigger question, why tell us about the few that can be?  Just because they can be identified, does not make them especially relevant.  Are there biological reasons to tell us about these specific compounds, if not, don't include.

Res: Thank you very much for the comments

Among the 566 molecular features extracted via LC-QTOF-MS analysis, a total of 34 secondary metabolites and one chlorophyll breakdown compounds were matched successfully. It is included in the manuscript.

Reviewer 2 Report

Manuscript Number: molecules-1038495

entitled: Antiparasitic Potential of Chromatographic Fractions of Nephrolepis biserrata and Liquid Chromatography–Quadrupole Time-of-Flight–Mass Spectrometry Analysis

The authors conducted an important data. This is an interesting paper. The present version of the manuscript is very well developed, the results contextualization, and the manuscript is well written. The data presented are new and relevant. The present version is much better; therefore, the manuscript is suitable for publication in the present form.

For the future, it is correct that in mass spectrometry analysis, regioisomers properties are not available; however, on GC or HPLC you should see the different retention time with exactly the same m/z.

Author Response

REVIEWER # 2

Comments and Suggestions for Authors

Manuscript Number: molecules-1038495

entitled: Antiparasitic Potential of Chromatographic Fractions of Nephrolepis biserrata and Liquid Chromatography–Quadrupole Time-of-Flight–Mass Spectrometry Analysis

The authors conducted an important data. This is an interesting paper. The present version of the manuscript is very well developed, the results contextualization, and the manuscript is well written. The data presented are new and relevant. The present version is much better; therefore, the manuscript is suitable for publication in the present form.

For the future, it is correct that in mass spectrometry analysis, regioisomers properties are not available; however, on GC or HPLC you should see the different retention time with exactly the same m/z.

Thank you so much for your comments.